# Active SLAM: A Review on Last Decade

**DOI:** 10.3390/s23198097

**Published:** 2023-09-27

**Authors:** Muhammad Farhan Ahmed, Khayyam Masood, Vincent Fremont, Isabelle Fantoni

**Affiliations:** 1Laboratoire des Sciences du Numérique de Nantes (LS2N), CNRS, Ecole Centrale de Nantes, 1 Rue de la Noë, 44300 Nantes, France; muhammad.ahmed@ec-nantes.fr (M.F.A.); isabelle.fantoni@ls2n.fr (I.F.); 2Capgemini Engineering, 4 Avenue Didier Daurat, 31700 Blagnac, France; khayyam.masood@capgemini.com

**Keywords:** SLAM, active SLAM, information theory, path planning, control theory

## Abstract

This article presents a comprehensive review of the Active Simultaneous Localization and Mapping (A-SLAM) research conducted over the past decade. It explores the formulation, applications, and methodologies employed in A-SLAM, particularly in trajectory generation and control-action selection, drawing on concepts from Information Theory (IT) and the Theory of Optimal Experimental Design (TOED). This review includes both qualitative and quantitative analyses of various approaches, deployment scenarios, configurations, path-planning methods, and utility functions within A-SLAM research. Furthermore, this article introduces a novel analysis of Active Collaborative SLAM (AC-SLAM), focusing on collaborative aspects within SLAM systems. It includes a thorough examination of collaborative parameters and approaches, supported by both qualitative and statistical assessments. This study also identifies limitations in the existing literature and suggests potential avenues for future research. This survey serves as a valuable resource for researchers seeking insights into A-SLAM methods and techniques, offering a current overview of A-SLAM formulation.

## 1. Introduction

Simultaneous localization and mapping (SLAM) is a set of approaches in which a robot autonomously localizes itself and simultaneously maps the environment while navigating through it. It can be subdivided into solving localization and mapping. Localization is a problem of estimating the pose of the robot with respect to the map, while mapping makes up the reconstruction of the environment with the help of visual, visual–inertial, and laser sensors on the robot. Modern SLAM approaches adopt a graphical approach (bipartite graph) where each node represents the robot or landmark pose and each edge represents a pose-to-pose or pose-to-landmark measurement. Consider a robot with state $x\in R^{2}$ describing its position and orientation (pose). The objective of the SLAM problem is to find the optimal state vector $x^{*}$, which minimizes the measurement error $e_{i}\left(x\right)$ weighted by the covariance matrix $\Omega_{i}\in R^{lxl}$, which encapsulates the measurement uncertainty in pose, and *l* is the dimension of the state vector, as shown in Equation (Equation 1). For a detailed discussion and review of SLAM methods, we can refer to [1,2,3,4,5]:(1)$$x^{*}=\arg\min_{x}\sum_{i}e_{i}^{T}\left(x\right)\Omega_{i}e_{i}\left(x\right)$$

SLAM algorithms are mostly passive, whereby the robot is controlled manually or goes toward predefined waypoints and the navigation or path-planning algorithm does not actively take part in robot motion or trajectory. A-SLAM, however, tries to solve the optimal exploration problem of the unknown environment by proposing a navigation strategy that generates future goal/target position actions that decrease the map and pose uncertainty, thus enabling a fully autonomous navigation and mapping SLAM system. We will look for further insight into A-SLAM in its designated Section 2. In Active Collaborative SLAM (AC-SLAM), multiple robots collaborate actively while performing SLAM. The application areas of A-SLAM and AC-SLAM include search and rescue [6], planetary observations [7], precision agriculture [8], autonomous navigation in crowded environments [9], underwater exploration [10,11,12], artificial intelligence [13], assistive robotics [14], and autonomous exploration [15].

The first implementation for an algorithm on A-SLAM was presented in [16], but the initial name was drafted in [17]. However, A-SLAM and its roots can be further traced back to the nineteen eighties from ideas coined by artificial intelligence and robotic exploration techniques [18]. Reviews on A-SLAM can be traced back to [19]. Within this article itself, A-SLAM is not the highlight of the research. Instead, the authors look at the whole topic of SLAM in its totality. Recently, the works of [20,21] provide some promising insight into A-SLAM formulation, methods, and future perspectives, as shown in Table 1. From Table 1, we can conclude that this article provides a comprehensive review of active and Active Collaborative SLAM research conducted mainly over the last decade on problem formulation, uncertainty quantification, optimal control, Deep Learning (DL), single- and multirobot analysis, limitations, and future perspectives. The contributions can summarized as follows: (1) This article discusses the A-SLAM formulation, methods, limitations, and future perspectives more comprehensively than most of the previous articles. (2) We provide a novel, extensive, qualitative and quantitative analysis of AC-SLAM. (3) We analyze research articles mostly from the last decade, which makes this review helpful for new researchers. Section 2 provides our motivation to review A-SLAM and an introduction to A-SLAM. In Section 2.1 and Section 2.2, we discuss A-SLAM formulation, its principal components, and how they are connected and related to each other. In Section 2.3, Section 2.4 and Section 2.5, we discuss the various techniques and application domains and provide qualitative analysis results. Section 2.7 presents our statistical analysis on robot-sensor-type usage, real robot usage, result types (simulation and analytical), the SLAM method adopted, the path-planning approach used, drive type, dataset usage, loop closure applicability, Robot Operating System (ROS) [22] usage, map type, and utility function usage. In Section 3, Section 3.1, Section 3.2, Section 3.3, Section 3.4 and Section 3.5, we present the AC-SLAM problem introduction, application domains, and qualitative and quantitative results, which quantify the collaboration architecture, collaboration parameters, environment usage, and utility functions apart from other parameters. In Section 4, we discuss the limitations of existing approaches and elaborate future research directions. Section 4.1 discusses the general limitations or open research problems. Section 4.2 and Section 4.3 highlight the potential limitations within the selected articles and future prospects for A-SLAM research. Finally, we conclude by summarizing this article and presenting our contributions in Section 5.

## 2. Introduction to Active SLAM (A-SLAM)

As described earlier, SLAM is a process in which a robot maps its environment and localizes itself to it. Referring to Figure 1, we observe that in SLAM, the front end handles perception tasks, which involve implementing methods in signal processing and computer-vision domains to compute the estimated relative pose of the robot and the landmarks (observed features). Data from the sensors, which are typically Light Detection and Ranging (Lidar), camera, and Inertial Measurement Unit (IMU) data, are processed by the front-end module, which computes feature extraction, data association, and feature classification and applies methods such as Iterative Closest Point (ICP) and loop closure to compute the estimated robot and landmarks pose with respect to the environment. The ICP is an iterative approach that computes the relative robot pose/transformation that optimizes/aligns the features and is used in scan-matching methods to map the environment. The back-end module is responsible for high-computational tasks involving Bundle Adjustment (B.A) and pose-graph optimization by using iterative solvers, e.g., Gauss–Newton [23] or Levenberg–Marquardt [24] algorithms, to solve the nonlinear optimization problem for an optimal estimate of the state vector $x^{*}$, as shown in Equation (Equation 1). The back-end module outputs the global map based on its sensor measurements by using Lidar/a camera and pose estimates of both the robot and landmarks.

A-SLAM deals with designing robot trajectories to minimize the uncertainty in the map representation and localization of the robot. The aim is to perform autonomous navigation and exploration of the environment without an external controller or human effort. A-SLAM can be referred to as an additional module or super set of SLAM systems that incorporates waypoints and trajectory planning and controller modules by using Information Theory (IT), control theory, and Reinforcement Learning (RL) methods to autonomously guide the robot toward its goal. During waypoint planning, A-SLAM chooses obstacle-free waypoints/points for suitable trajectory. Trajectory planning integrates these points with time and generates a trajectory for the robot to follow. The controller sends actuator commands to the robot to follow the desired trajectory and reach the goal position. We will discuss these components comprehensively in Section 2.2 and discuss the applications of IT and RL methods in Section 2.3 and Section 2.4, respectively.

In SLAM, environment exploration (to obtain better knowledge of the environment) and exploitation (to revisit already-traversed areas for loop closure) are maximized for better map estimation and localization. As a consequence, we have to perform a trade-off between exploration and exploitation as the prior requires maximum coverage of the environment and the latter requires the robot to revisit previously explored areas. These two tasks may not always be applied simultaneously for a robot to perform autonomous navigation. The robot might have to solve the exploration–exploitation dilemma by switching between these two tasks.

In Section 2.1 and Section 2.2, we provide the basic A-SLAM formulation and its main components along with their brief definitions and functions in the A-SLAM pipeline.

### 2.1. A-SLAM Formulation

A-SLAM is formulated in a scenario where the robot has to navigate in a partially observable/unknown environment by selecting a series of future actions in the presence of noisy sensor measurements that reduce its state and map uncertainties with respect to the environment. Such a scenario can be modeled as an instance of the Partially Observable Markov Decision Process (POMDP), as discussed in [21,25,26]. The POMDP is defined as a seven tuple $\left(X,A,O,T,\rho_{o},\beta ,\gamma \right)$, where X∈R represents the robot state space and is represented as the current state x∈X and the next state $x^{{}^{\prime }}\in X$; A∈R is the action space and can be expressed as a∈A; *O* represents the observations where o∈O; *T* is the state-transition function between an action (*a*), present state (*x*), and next state ($x^{{}^{\prime }}$); *T* accounts for the robot control uncertainty in reaching the new state $x^{{}^{\prime }}$; $\rho_{o}$ accounts for sensing uncertainty; β is the reward associated with the action taken in state *x*; and γ∈(0,1) takes into account the discount factor ensuring a finite reward even if the planning task has an infinite horizon. Both *T* and $\rho_{o}$ can be expressed by using conditional probabilities as Equations (Equation 2) and (Equation 3):(2)$$T\left(x,a,x^{{}^{\prime }}\right)=p\left(x^{{}^{\prime }}|x,a\right)$$
(3)$$\rho_{o}\left(x,a,o\right)=p\left(o|x^{{}^{\prime }},a\right)$$

We can consider a scenario where the robot is in state *x* and takes an action *a* to move to $x^{’}$. This action uncertainty is modeled by *T* with an associated reward modeled by β, and then it takes an observation *o* from its sensors that may not be precise in their measurements, and this sensor uncertainty is modeled by $\rho_{o}$. The robot’s goal is to choose the optimal policy $\alpha^{*}$ that maximizes the associated expected reward (E) for each state–action pair, and it can be modeled as Equation (Equation 4):(4)$$\alpha^{*}=\underset{t}{\mathrm{argmax}}\sum_{t=0}^{\infty }E\gamma^{t}\beta \left(x_{t},a_{t}\right)$$
where $x_{t}$, $a_{t}$, and $\gamma^{t}$ are the state, action, and discount factor evolution at time *t*, respectively. Although the POMDP formulation of A-SLAM is the most widely used approach, it is considered computationally expensive as it considers planning and decision making under uncertainty. For computational convenience, A-SLAM formulation is divided into three main submodules which identify the potential goal positions/waypoints, compute the cost to reach them, and then select actions based on utility criterion, which decreases the map uncertainty and increases the robot’s localization. We will discuss these submodules briefly in Section 2.2.

### 2.2. A-SLAM Components

To deal with the computational complexity of A-SLAM, it is divided into three main submodules, as depicted in Figure 2. The robot initially identifies potential goal positions to explore or exploit its current estimate of the map. The map represents the environment perceived by the robot by using its onboard sensors, and it may be classified as (1) topological maps, which use a graphical representation of the environment and provide a simplified topological representation; (2) metric maps, which provide environment information in the form of a sparse set of information points (landmarks) or full 3D representation of the environment (point cloud); and (3) semantic maps, which provide only segmented information about environment objects (like static obstacles) to the robot. Interested readers are directed to [1,19] for a detailed discussion on mapping approaches. Once the robot has a map of its environment by using any of the above approaches, it searches for potential target/goal locations to explore. One of the most widely used methods is frontier-based exploration initially used by [27], where the frontier is the border between the known and unknown map locations. Figure 3 shows frontiers detected by using Lidar measurements on the occupancy grid map in a simulation environment. Using frontier-based exploration has the advantage that all the environment may be covered, but no exploitation task (revisiting already-visited areas for loop closure) is performed, which affects the robot’s map estimate; we will discuss the application of this approach in Section 2.3.2.

Once the goal position is identified, the next step is to compute the cost or utility function to that position based on the reward value of the optimal action selected from a set of all possible actions according to Equation (Equation 4). Ideally, this utility function should consider a full joint-probability distribution of map and robot poses, but this method is computationally expensive. Since we have a probabilistic estimation of both the robot and map, we can treat them as random variables with associated uncertainty in their estimation. The two most common approaches used in the quantification of this uncertainty are Information Theory (IT), initially coined by Shannon in 1949, and the Theory of Optimal Experimental Design (TOED) [29].

In IT, entropy measures the amount of uncertainty associated with a random variable or random quantity. Higher entropy leads to less information gain and vice versa. Formally, it is defined for a random variable *X* as H(X), as shown in Equation (Equation 5):(5)$$H\left(X\right)=-\sum_{x\in X}p\left(x\right)\log_{2}p\left(x\right)$$

Since both the robot pose and the map are estimated as a multivariate Gaussian, the authors of [30] formulate the Shannon’s entropy of the robot pose as in Equation (Equation 6), where *n* is the dimension of the robot pose vector and $\Omega \in R^{nxn}$ is the covariance matrix. The map entropy is defined as Equation (Equation 7), where the map *M* is represented as an occupancy grid and each cell $m_{i,j}$ is associated with a Bernoulli distribution $P(m_{i,j})$. The objective is to reduce both the robot pose and map entropy. Relative entropy is also be used as a utility function that measures the probability distribution along with its deviation from its mean. This relative entropy is measured as the Kullback–Leibler divergence (KLD). The KLD for two discrete distributions *A* and *B* on probability space *X* can be defined as Equation (Equation 8):(6)$$H\left[p\left(x\right)\right]=\frac{n}{2}(1+\log\left(2\pi \right)+\frac{1}{2}\log\left(det\Omega \right)$$
(7)$$H\left[p\left(M\right)\right]=-\sum_{i,j}(p\left(m_{i,j}\right)log\left(p\left(m_{i,j}\right)\right)+\left(1-p\left(m_{i,j}\right)\right)log\left(1-p\left(m_{i,j}\right)\right)$$
(8)$$D_{KL}\left(A|B\right)=\sum_{x\in X}A\left(x\right)\log\frac{A(x)}{B(x)}$$

When considering information-driven utility functions, entropy or KLD can be used as a metric to target binary probabilities in the grid map (occupancy grid map). Alternatively, if we consider task-driven utility functions where the uncertainty metric is evaluated by reasoning over the propagation of uncertainty in the pose-graph SLAM covariance matrix, we can quantify the uncertainty in the task space. TOED provides many optimal criteria, which transform the mapping of the covariance matrix to a scalar value. Hence, by using TOED, the priority of a set of actions for A-SLAM is based on the amount of covariance in the joint posterior. Less covariance contributes to a higher weight of the action set. The “optimality criterion” used in TOED can be defined for a covariance matrix $\Omega \in R^{nxn}$ and eigenvalues $\zeta_{n}$ as (1) A-optimality, which deals with the minimization of the average variance, as shown in Equation (Equation 9); (2) D-optimality, which deals with minimizing the volume of the covariance ellipsoid and is defined in Equation (Equation 10); and (3) E-optimality, which intends to minimize the maximum eigenvalue and is expressed in Equation (Equation 11):(9)$$A-Opt\overset{\Delta }{=}\frac{1}{n}\left(\sum_{k=1}^{n}\zeta_{k}\right)$$
(10)$$D-Opt\overset{\Delta }{=}exp\left(\frac{1}{n}\sum_{k=1}^{n}log\left(\zeta_{k}\right)\right)$$
(11)$$E-Opt\overset{\Delta }{=}min_{1\leq i\leq n}\left(\zeta_{i}\right)$$

TOED approaches require both the robot pose and map uncertainties to be represented as a covariance matrix and may be computationally expensive, especially in landmark-based SLAM where its size increases as new landmarks are discovered. Hence, IT-based approaches are preferred over TOED. We will discuss the application of these approaches in Section 2.3.

Once the goal positions and utility/cost to reach these positions have been identified, the next step is to execute the optimal action, which eventually moves/guides the robot to the goal position. Three approaches are commonly deployed:Probabilistic Road Map (PRM) approaches discretize the environment representation and formulate a network graph representing the possible paths for the robot to select to reach the goal position. These approaches work in a heuristic manner and may not give the optimal path; additionally, the robot model is not incorporated in the planning phase, which may result in unexpected movements. Rapidly exploring Random Trees (RRT) [31], D* [32], and A* [33] are the widely used PRM methods. We identify these methods as geometric approaches, and in Section 2.3, we discuss the application of these methods.Linear Quadratic Regulator (LQR) and Model Predictive Control (MPC) formulate the robot path-planning problem as an Optimal Control Problem (OCP) and are used to compute the robot trajectory over a finite time horizon in a continuous-planning domain. Consider a robot with the state-transition equation given by x(k+1)=f(x(k),u(k)), where *x*, *u*, and *k* are the state, control, and time, respectively. The MPC controller finds the optimal control action $u^{*}\left(x\left(k\right)\right)$ for a finite horizon *N*, as shown in Equation (Equation 12), which minimizes the relative error between the desired state $x^{r}$ and desired control effort $u^{r}$, weighted by matrices *Q* and *P* for the penalization of state and control errors, respectively, as shown in Equation (Equation 13). The MPC is formulated as minimizing the objective function $J_{N}$ as defined in (Equation 14), which takes into account the costs related to control effort and robot state evolution over the entire prediction horizon. MPC provides an optimal trajectory incorporating the robot state model and control and state constraints, making it suitable for path planning in dynamic environments:
(12)$$u^{*}\left(x\left(k\right)\right):=\left(u^{*}\left(k\right),u^{*}\left(k+1\right),....u^{*}\left(k+N-1\right),\right)$$
(13)$$l\left(x,u\right)={∥x_{u}-x^{r}∥}_{Q}^{2}+{∥u-u^{r}∥}_{R}^{2}$$
(14)$$\underset{u}{\mathrm{minimize}}J_{N}\left(x_{0},u\right)=\sum_{k=0}^{N-1}l\left(\left(x_{u}\left(k\right),u\left(k\right)\right)\right)$$Reinforcement Learning (RL) is modeled as a Markov Decision Process (MDP) where an agent at state *s* interacts with the environment by performing an action *a* and receiving a reward *r*. The objective is to find a good policy π(s,a) which maximizes the aggregation of the rewards in the long run following a value function $V_{\pi }\left(s_{t_{0}}\right)$, as shown in Equation (Equation 15), that maximizes the expected reward attained by the agent, weighted by the discount factor $\gamma^{t}\in \left[0,1\right]$. In the case of visual A-SLAM, the policy may be to move the robot to more feature-rich positions to maximize the reward (observed features). Deep Reinforcement Learning (DRL) replaces the agent with a deep neural network that parameterizes the policy π with some weighting parameter θ and is given as $\pi_{\theta }\left(s,a\right)$ to maximize the future rewards of each state–action pair during the evolution of the robot trajectory. We will further discuss the application of these approaches in Section 2.4:
(15)$$V_{\pi }\left(s_{t_{0}}\right)=\sum_{t=t_{0}}^{\infty }\gamma^{t}r\left(s_{t},\pi \left(s_{t},a_{t}\right)\right)$$

The choice of selecting a suitable waypoint candidate is weighted by using IT and TOED, as discussed in previous sections. In these methods, information gain or entropy minimization between the map and robot path guides the decision for the selection of these future waypoint candidates. To generate a trajectory or a set of actions for these future waypoint candidates, two main methods are adopted, namely geometric and dynamic approaches, respectively. These methods involve the usage of traditional path planners along with the DRL and nonlinear optimal control techniques. In Section 2.3, Section 2.4 and Section 2.5, we will discuss these two methods and their utilization in the research articles that are a part of this survey.

### 2.3. Geometric Approaches

These methods describe A-SLAM as a task for the robot whereby it must choose the optimal path and trajectory while reducing its poses and mapping uncertainty for efficient SLAM to autonomously navigate an unknown environment. The exploration space is discretized with finite random waypoints and frontier-based exploration along with traditional path planners like RRT*, D*, and A*, which are deployed with IT- and TOED-based approaches including entropy, KLD, and uncertainty metrics reduction. We can further classify the application of these approaches as follows in the sections below.

#### 2.3.1. IT-Based Approaches

The authors of [34] address the joint-entropy minimization exploration problem and propose two modified versions of RRT* [31] called dRRT* and eRRT*. dRRT* uses distance, while eRRT* uses entropy change per distance traveled as the cost function. It is further debated that map entropy has a strong relationship with coverage and path entropy has a relationship with map quality (as better localization produces a better map). Hence, actions are computed in terms of the joint-entropy change per distance traveled. The simulation results proved that a combination of both of these approaches provides the best path-planning strategy. An interesting comparison between IT approaches is given in [35], where particle filters are used as the back end of A-SLAM and frontier-based exploration (a frontier is a boundary between the visited and unexplored areas) [27] is deployed to select future candidate target positions. A comparison of these three methods used for solving the exploration problem and evaluating the information is discussed in the relevant sections below:Joint entropy: The information gained at the target is evaluated by using the entropy of both the robot trajectory and map carried by each particle weighted by each trajectory’s importance weight. The best exploration target is selected, which maximizes the joint-entropy reduction and hence corresponds to higher information gain.Expected Map Mean: An expected mean can be defined as the mathematical expectation of the map hypotheses of a particle set. The expected map mean can be applied to detect already-traversed loops on the map. Since the computation of the gain is developing, the complexity of this method increases.Expected information from a policy: Kullback–Leibler divergence [36] is used to drive an upper bound on the divergence between the true posterior and the approximated pose belief. Apart from the information consistency of the particle filter, this method also considers the information loss due to inconsistent mapping.

It was concluded by using simulation results on various datasets, that most of these approaches were not able to properly address the probabilistic aspects of the problem and are most likely to fail because of a high computational cost and the map-grid resolution dependency on performance.

The authors of [37] use an exploration space represented by primitive geometric shapes, and an entropy reduction over the map features is computed. They use an entropy metric based on Laplacian approximation and compute a unified quantification of exploration and exploitation gains. An efficient sampling-based path planner is used based on a Probabilistic Road Map approach, having a cost function that reduces the control cost (distance) and collision penalty between targets. The simulation results compared to the traditional grid-map frontier exploration show a significant reduction in position, orientation, and exploration errors. Future improvements include expanding to an active visual SLAM framework.

When considering topometric graphs and a less computationally expensive solution, we can refer to the approach adopted by [38], which considers a scenario where we have many prior topometric subgraphs and the robot does not know its initial position. A novel open-source framework is proposed that uses active localization and active mapping. A submap-joining approach is defined, which switches between active localization and mapping. Active localization uses the maximum likelihood estimation to compute a motion policy, which reduces the computational complexity of this method.

#### 2.3.2. Frontier-Based Exploration

Frontiers are boundaries between the explored and unexplored space. Formally, we can describe frontiers as a set of unknown points that each have at least one known space neighbor. The work presented by [39] formulates a hybrid control-switching exploration method of particle filter SLAM as the back end. It uses a frontier-based exploration method with A* [33] as a global planner and the Dynamic Window Approach (DWA) reactive algorithm as a local planner. Within the occupancy grid map, each frontier is segmented, a trajectory is planned for each segment, and the trajectory with the highest map-segment covariance is selected from the global-cost map. The work presented in [9] deals with dynamic environments with multiple ground robots and uses frontier exploration for autonomous exploration with graph-based SLAM (iSAM) [40] optimization as the SLAM back end. Dijkstra’s algorithm-based local planner is used. Finally, a utility function based on Shannon’s and Renyi entropy is used for the computation of the utility of paths. Future work proposes to integrate a camera and use image-feature scan matching for obstacle avoidance.

#### 2.3.3. Path-Planning Optimization

The method proposed by [10] exploits the relationship between the graphical model and sparse matrix factorization of graphical SLAM. It proposes the ordering of variables and a subtree-catching scheme to facilitate the fast computation of optimized candidate paths weighted by the belief changes between them. The horizon selection criteria are based on the author’s previous work utilizing an extended information filter (EIF) and Gauss–Newton (GN) prediction. The proposed solution is implemented in a Hovering Autonomous Underwater Vehicle (HAUV) with pose-graph SLAM. The work presented in [12] deals with a similar volumetric exploration in an underwater environment with a multibeam sonar. For efficient path planning, the revisit actions are selected depending on the pose uncertainty and sensor-information gain.

The authors of [41] used an interesting approach that addresses the path-planning task as D* [32] with negative edge weights to compute the shortest path in the case of a change in localization. This exploration method is highly effective in dynamic environments with changing obstacles and localization. When dealing with noisy sensor measurements, an interesting approach is adopted by [42], which proposes a system that makes use of a multihypothesis state and map estimates based on noisy or insufficient sensor information. This method uses the local contours for efficient multihypothesis path planning and incorporates loop closure.

#### 2.3.4. Optimization in Robot Trajectory

The method proposed in [43] integrates A-SLAM with Ekman’s exploration algorithm [44] to optimize the robot trajectory by leveraging only the global waypoints where loop closure appears, and then the exploration canceling criterion is sent to the SLAM back end (based on the information filter [45]). The exploration canceling criterion depends on the magnitude of information gain from the filter, loop-closure detection, and the number of states without an update. If these criteria are met, the A-SLAM causes the exploration algorithm to stop and guides the robot to close the loop. We must note that in this approach, A-SLAM is separated from the route-planning and -exploration process, which is managed by the information filter. In a similar approach presented by [46], this study assumes that some prior map information about the environment is available as a topological map. Then, A-SLAM exploits this map information for active loop closure. The proposed method calculates an optimal global plan as a solution to the Chinese Postman Problem (CPP) [47] and an online algorithm that computes the maximum likelihood estimate (MLE) by using nonlinear optimization, which computes the optimized graph with respect to the prior map and explored map. The D-optimality criterion is used to represent the robot localization uncertainty while the work presented by [7] incorporates active path planning with salient features (features with a high entropy value) and ICP-based feature matching [48]. The triggering condition of A-SLAM is based on an active feature revisit, and the path with the maximum utility score is chosen based on its length and map data.

#### 2.3.5. Optimal Policy Selection

The definition and comparison presented in [49] formulate A-SLAM as a task of choosing a single or multiple policy type for robot trajectories, which minimizes an objective function that comprises a reduction in the expected costs of robot uncertainty, energy consumption, and navigation time among other factors. An optimality criterion by definition quantifies the improvement in the actions taken by the robot to improve the localization accuracy and navigation time. A comparison between D-optimality (proportional to the determinant of the covariance matrix), A-optimality (proportional to the trace of the covariance matrix), and joint entropy is performed, and it is concluded that the D-optimality criterion is more appropriate for providing useful information about the robot’s uncertainty contrary to A-optimality. The authors of [50] proved numerically that by using differential representations to propagate the spacial uncertainty, monotonicity is preserved for all the optimality criteria A-opt, D-opt, and E-opt (the largest eigenvalue of the covariance matrix). In absolute representations using only unit quaternions, the monotonicity is preserved only in D-optimality and Shannon’s entropy. In a similar comparison, the work presented in [51] concludes that A-Opt and E-opt criteria do not hold monotonicity in dead reckoning scenarios. It is proved by using simulations with a differential drive robot that the D-opt criterion, under a linearized odometry method, holds monotonicity.

### 2.4. Dynamic Approaches

Instead of using traditional path planners like A*, D*, and RRT, these methods formulate the A-SLAM as a problem with selecting a series of control inputs to generate a collision-free trajectory and cover as much area as possible while minimizing the state-estimation uncertainty and thus improve the localization and mapping of the environment. The planning and action spaces are now continuous (contrary to being discrete in geometry-based methods) and local optimal trajectories are computed. For the selection of optimal goal positions, similar approaches to the geometric approach methods in Section 2.3 are used with the exception that now the future candidate trajectories are computed by using robot models, potential information fields, and control theory. A Linear Quadratic Regulator (LQR), Model Predictive Control (MPC) [52], the Markov Decision Process [53], or Reinforcement Learning (RL) [54] are used to choose the optimal future trajectories/set of trajectories via metrics that balance the need for exploring new areas and exploiting already-visited areas for loop closure.

The method used by [55] uses Reinforcement Learning in the path planner to acquire a vehicle model by incorporating a 3D controller. The 3D controller can be simplified to one 2D controller for forward and backward motion and one 1D controller for path planning that has an objective function that maximizes the map reliability and exploration zone. Therefore, the planner has an objective function that maximizes the accumulated reward for each state–action pair by using the “learning from experience approach”. It is shown through simulations that a nonholonomic vehicle learns the virtual wall-following behavior. A similar approach presented in [13] uses fully convolutional residual networks to recognize the obstacles and obtain a depth image. The path-planning algorithm is based on DRL.

An active localization solution where only the rotational movement of the robot is controlled in a position-tracking problem is presented by [56]. The Adaptive Monte Carlo Localization (AMCL) particle cloud is used as the input, and robot-control commands are sent to its sensors as the output. The proposed solution involves the spectral clustering of the point cloud, building a compound map from each particle cluster, and selecting the most informative cell. The active localization is triggered when the robot has more than one cluster in its uncertainty estimate. The future improvements include more cells for efficient hypotheses estimation and integrating this approach into the SLAM front end. In an interesting approach by [57], the saccade movement of bionic eyes (rapid movement of the center of one’s gaze within the visual field) is controlled. To leverage more features from the environment, an autonomous control strategy inspired by the human vision system is incorporated. The A-SLAM system involves two threads (parallel processes), a control thread, and a tracking thread. The control thread controls the bionic eyes’ movement to Oriented FAST and Rotated Brief (ORB) feature-rich positions while the tracking thread tracks the eye motion by selecting the feature-rich keyframes.

### 2.5. Hybrid Approaches

These methods use the geometry and dynamic-based methods mentioned in Section 2.3 and Section 2.4 incorporating frontier-based exploration, Information Theory, and Model Predictive Control (MPC) to solve the A-SLAM problem.

The approach used by the authors of [58] presents an open-source multilayer A-SLAM approach where the first layer selects the informative (utility criterion based on Shannon’s entropy [59]) goal locations (frontier points) and generates paths to these locations while the second and third layers actively replan the path based on the updated occupancy grid map. Nonlinear MPC [60] is applied for local path execution with the objective function based on minimizing the distance to the target and controlling the effort and cost of being close to a nearby obstacle. One issue with this approach is that sometimes the robot stops and starts the replanning phase of local paths. Future works should involve adding dynamic obstacles and the usage of aerial robots.

An interesting approach mentioned in [8,61] presents a solution based on Model Predictive Control (MPC) to solve the area coverage and uncertainty reduction in A-SLAM. An MPC control-switching mechanism is formulated, and SLAM uncertainty reduction is treated as a graph topology problem and planned as a constrained nonlinear least-squares problem. Using convex relaxation, the SLAM uncertainty is reduced by a convex optimization method. The area-coverage task is solved via the sequential quadratic programming method, and Linear SLAM is used for submap joining.

### 2.6. Reasoning over Spectral Graph Connectivity

Recently, in the works of [62,63], the authors exploit the graph SLAM connectivity and pose it as an estimation-over-graph (EoG) problem, where each node (state vector) and the vertex (measurement) connectivity is strongly related to the SLAM estimation reliability. By exploiting the spectral graph theory, which deals with the eigenvalues, Laplacian, and degree matrix of the associated SLAM information matrix and graph connectivity, the authors state that (1) the graph Laplacian is related to the SLAM information matrix and (2) the number of Weighted Spanning Trees (WST) is directly related to the estimation accuracy of the graph SLAM.

The authors of [64,65,66] extend [63] by debating that the maximum number of WST is directly related to the maximum likelihood (ML) estimate of the underlying graph SLAM problem formulated over lie algebra [67]. Instead of computing the D-optimality criterion defined in Equation (Equation 10) over the entire SLAM sparse-information matrix, it is computed over the weighted graph Laplacian where each vertex is weighted by using D-optimality, and it is proven that the maximum number of WST of this weighted graph Laplacian is directly related to the underlying pose-graph uncertainty.

Real robot experiments on both the Lidar and visual SLAM backbends prove the efficiency and robustness of reasoning the uncertainty over the SLAM-graph connectivity.

### 2.7. Statistical Analysis on A-SLAM

Table 2 summarizes the sensor types, SLAM methods, path-planning approaches, and publication years of the selected articles. We can further debate that in most A-SLAM methods, (i) Lidar (62%), RGB (28%), and RGBD (19%) camera sensors are mostly used as the main input data source to extract the point cloud and image features/correspondences. (ii) Extended Kalman Filter (EKF)- or particle-filter-based SLAM methods are mostly used (54%) as compared to pose-graph- or graph-based SLAM methods (45%) along with g2o [68], incremental smoothing and mapping (iSAM) [40], and Georgia Tech Smoothing and Mapping (GTSAM) [69] as the main graph-optimization frameworks/libraries. Hence, we can conclude that although modern graph SLAM approaches are more robust, efficient, and consume less memory as compared to filter-based methods, their usage in A-SLAM is discouraged in comparison. (iii) Path-planning algorithms that discretize the search space including A*, D*, and sampling-based approaches like RRT, RRT*, and Dijkstra’s are used 19% and 25% of the time alongside DRL-based approaches while continuous space-planning algorithms, which incorporate robot kinematics models like MPC, Timed Elastic Band (TEB) [70], and Dynamic Window-Based (DWA) [71] approaches are only used 11% of the time.

Table 3 summarizes the robots and their drive types (locomotion mechanisms), dataset usage, loop closure, and ROS [22] implementations used in the selected A-SLAM articles. The information can be summarized as the following: (i) Most implementations for experimental validation use about 80% of the commercially available ground robots. This reliance motivates their usage for research purposes. (ii) Drive mechanisms that define the kinematic model and robot movement in the environment can be characterized into (a) differential drive (two fixed wheels and one caster wheel); (b) skid-steering four-wheel drive (four fixed wheels); (c) Ackerman drive (car-like robots with two fixed wheels and two steerable wheels); (d) traction drive (a chain structure used for movement instead of wheels); and (e) omnidirectional drive (which uses special wheels for holonomic motion), where the position and orientation of the robot can be controlled independently. We can deduce that most approaches use physical robots (55%), differential dive (25%), and skid-steering drive mechanisms (20%). The differential drive is preferred because of the simple kinematic model as compared to Ackerman and omnidirectional drives. (iii) Dataset usage is limited to only 20%. This reduced usage of available open-source datasets motivates the fact that, in A-SLAM, it is difficult to provide a dataset since the control commands are also incorporated, which makes the usage impractical in different environments. (iv) Loop closure is incorporated in 51% of implementations. Loop closure greatly enhances the SLAM efficiency, and its usage should be encouraged. (v) ROS, which is a popular open-source software platform used for educational purposes, is used only 45% of the time for most implementations, and its usage should be encouraged. (vi) Occupancy grid maps (53.1%) and topological maps (37.5%) are mostly used as compared to point clouds for environment representation because of their simplistic nature and computational requirements as compared to dense point-cloud maps. (vii) For the computation of the utility function, entropy (28.1%) and D-optimality (21.8%) are mostly used along with FD (15.6%).

In Figure 4, the per annum selection of A-SLAM articles and the usage of ROS is depicted. Referring to Figure 4a, the per annum percentage of the selected articles is shown, depicting the dataset used in this survey. We can observe that almost 69% of the articles are selected from the last seven years, providing the latest information on A-SLAM research. In Figure 4b, although ROS is a popular environment for robots, its application should be encouraged as it is deployed only in 39% of A-SLAM solutions.

Figure 5 shows the percentage of real robots used and the annual distribution of articles showing results from using actual robots in their experiments. From Figure 5a, we can observe that only 47% of articles use real robots for experimental validation while 53% do not. Although this is a narrow gap, still we encourage real robot usage to motivate real-world applications. Figure 5b shows the annual distribution of articles using real robots in their experiments. We can further debate that up until 2015, only 20% of articles used real robots, while from 2016 to 2023 (last 8 years inclusive), the usage increased to 54%, which is very encouraging and shows the real-world applications of the proposed methods.

Figure 6 elaborates on the percentage of simulation results and their annual distribution. In Figure 6a, we can observe that 85% of the articles provide simulation results as compared to that of 15% which do not. This large percentage of simulation results indicates the effectiveness of the proposed algorithms in the simulation environments. From Figure 6b, we observe that from 2011 to 2015, almost 90% of the articles provided simulation results, and from 2016 to 2023 (inclusive), around 87% provided the same. Hence, we can observe a high percentage of simulation results, which promises the high applicability of the proposed solutions in simulation environments.

Figure 7 presents the percentage of analytical results and their yearly disposition over the last 12 years. From Figure 7a, we can infer that 88% of the articles give analytical results while 12% provide either simulation or real robot experimental results. This large percentage of analytical results is highly beneficial as it presents the necessary mathematical foundations of the proposed algorithm. From Figure 7b, we observe that from 2011 to 2015, almost 90% of the articles provided analytical results, while from 2016 to 2023 (inclusive), the percentage was around 87%.

## 3. Active Collaborative SLAM (AC-SLAM)

In collaborative SLAM (C-SLAM), multiple robots interchange information to improve their localization estimation and map accuracy to achieve some high-level tasks such as exploration. This collaboration raises some challenges regarding the usage of computational resources, communication resources, and the ability to recover from network failure. The exchanged information should detect inter-robot correspondences and estimate trajectories while estimating the state of the robots. These inter-robot interactions should not compromise the available computational and memory resources required by other SLAM processes (loop closure and visual-feature correspondences). The robots should efficiently utilize the limited communication bandwidth and range.

In AC-SLAM, the TOED-, IT-, and control-theory-based approaches using A-SLAM mentioned in Section 2.3, Section 2.4 and Section 2.5 are also applicable with additional constraints of managing the communication and parameter exchange between robots as mentioned above. Table 4 presents the collaboration parameters exchanged between the AC-SLAM robots. These parameters are entropy, KLD, localization info, visual features, and frontier points. In addition to these parameters, AC-SLAM parameters may include (a) the parameters presented by the authors of [86,87], incorporating the multirobot constraints induced by adding the future robot paths while minimizing the optimal control function (which takes into account the future steps and observations) and minimizing the robot state and map uncertainty and adding them into the belief space (assumed to be Gaussian); (b) parameters relating to exploration and relocalization (to gather at a predefined meeting position) phase of robots as described by [88]; (c) 3D mapping info (OctoMap) used by the authors of [89]; and (d) path and map entropy info, as used in [90], and relative entropy, as mentioned in [91].

### 3.1. Network Topology of AC-SLAM

A network topology (communication topology) describes how different robots/nodes communicate and exchange data with each other and with a central computer/server. Figure 8 summarizes different communication topologies. This communication may be centralized, decentralized, or distributed. In a centralized communication network as presented by the authors of [86,87,91,94,95,97], all the communication between the nodes is routed through the central server. If the central server becomes unavailable/out of service/out of range, then communication is broken, which makes this topology highly vulnerable to communication loss in the case of server failure. Table 4 shows that about 50% of the selected articles use this type of topology. A decentralized network may be considered as a subset of a centralized one where a common central server routes communication with different subcentralized networks, hence making the entire network highly dependent on it. In a distributed network, as shown in [88,89,90,92,93,96,98,99,100,101], each node communicates with each other without need for a central server. This topology is highly reliable and is used by 62% of the articles, referring to Table 4. As communication is not rallied through any central server and each node handles onboard computation and network communication, this topology is less vulnerable to communication loss. Since there is dense communication between nodes, managing the communication bandwidth, task allocation, and data packet reduction considerations are the parameters that need to be optimized, as presented in the work of [102]. Typical application scenarios include collaborative localization [87,92,94,97], exploration and exploitation (revisiting already-explored areas for loop closure) [15,96], and collaborative trajectory planning/trajectory optimization [90,91,95]. In the following sections, we discuss these application scenarios in the selected literature.

### 3.2. Collaborative Localization

In these methods, the robots switch their states (tasks) between self/independent localization and assistive localization to other robots. The method proposed by the authors of [92] presents a novel centralized method in which a DRL-based task-allocation algorithm is used to assist agents in a relative observation task. To learn the correspondence between the quality function (Q) and state–action pair, a novel multiagent deep Q network is deployed. Each agent can choose to perform its independent ORB-SLAM [103] or localize other agents. The reward function incorporates the influence of the other agent’s transition error in decision making. The observation function is derived from ORB-SLAM and consists of map points, keyframes, and loop-closure-detection components. To compute the relative observation between agents, the nonlinear optimization problem is solved by using the Gauss–Newton algorithm to estimate the pose of the target agent. The large associated computational cost of this method lacks real-time application and thus a distributed learning approach is proposed in the future.

The method described in [87] presents the mutirobot state estimation problem as a belief-space-spanning problem by exploiting the POMDP nature of A-SLAM. The authors measured the robot belief as the probability distribution of its state from the entire group and mapped environment. The proposed active-localization method can guide each robot by using Maximum A Posteriori (MAP) estimation of future waypoints and reduce its uncertainty by reobserving areas only observed by other robots. The proposed objective function takes into account the evolution of predicted measurements over the planning horizon and the trace of the covariance matrix associated with the robot-pose uncertainty. In an interesting approach, the method presented in [97] uses multiple humanoid robots, where each robot has two working modes, independent and collaborative. Each robot has two threads running simultaneously: (a) the motion thread and (b) the listening thread. With the motion thread, it will navigate the environment via the trajectory computed by the organizer (central server) by using a D* path planner and a control strategy based on DRL and a greedy algorithm. It also uploads its pose periodically to the organizer. With the listening thread, it will receive its updated pose from the organizer (via ORB-SLAM) and may receive the command to help other robots in the vicinity improve their localization by using a chained localization method. With this method, each robot’s localization is improved by its preceding robot, and its covariance is updated depending on the measurement error between the two robots. In [94], the authors propose a method to rectify the weak connections in the target robot’s pose graphs by the host robot. These weak connections are identified when the covariance increases to a certain threshold and is communicated as a result of the Edge Selection Problem (ESP) [63], whereby the host robots generate trajectories toward them by using RRT to decrease uncertainty and improve their localization. This method uses continuous refinement along with D-optimality criterion to collaboratively plan trajectories that reduce pose uncertainty in pose-graph-based SLAM. A bidding strategy is defined, which selects the winning robot based on the least computational cost, feasible trajectory, and resource-friendly criterion.

### 3.3. Exploration and Exploitation Tasks

As mentioned earlier, in A-SLAM, we need to balance exploration (maximizing the explored area) and exploitation (revisiting already-explored areas for loop closure). This balance is important to achieve good robot- and landmark-pose estimations and achieve better localization and mapping. The authors of [96] describe a centralized exploration problem by using frontier-based exploration and an efficiency-optimization problem where the information gain and localization efficiency is maximized while navigation cost toward the frontier is penalized. During the exploration phase, a global optimization strategy is proposed, which divides the exploration task equally among robots. Sometime during the exploration phase, the robot’s localization efficiency drops a certain threshold and it switches to the relocalization phase. During the relocalization (exploitation) phase, each robot is guided toward a known landmark or another robot with less localization uncertainty. An adapted threshold criterion is defined, which is adjusted by the robots to escape the exploring and exploitation loop if they get stuck. To manage the limited communication bandwidth (because of a centralized architecture), a rendezvous method is proposed, which relocates the robots to a predefined position if they get out of the communication range. The future work proposed involves using distributed control schemes.

The method described in [15] formulates the problem in topological, geometrical space (the environment which is represented by primitive geometric shapes). Initially, the robots are assigned target positions, and exploration is based on the frontier method and utilizing a switching cost function that takes into consideration the discovery of the target area of a robot by another member of the swarm. When the target is inside the robot’s disjoint explored subspace, the cost function switches from a frontier to a distance-based navigation function to guide the robot toward the goal frontier.

### 3.4. Trajectory Planning

In these methods, the path entropy is optimized to select the most informative path to collectively plan trajectories that reduce the localization and map uncertainties. In the approach formulated in [90], the study presents a decentralized method for a long-planning horizon of actions for exploration and maintains estimation uncertainties at a certain threshold. The active path planner uses a modified version of RRT* in which (a) the nonfusible nodes are filtered out because a nonholonomic robot is used and (b) the action is chosen that best minimizes the entropy change per distance traveled. The authors performed entropy estimation as a two-stage process. At first, the entropy in short horizons is computed by using square root information filter (SRIF) updates and that of the short horizon is computed considering a reduction in the loop closures in the robot paths. The main advantage of this approach is that it maintains good pose estimation and encourages loop-closure trajectories. An interesting solution is given by a similar approach to the method proposed by [91] using a relative entropy (RE)-optimization method which integrates motion planning with robot localization and selects trajectories that minimize the localization error and associated uncertainty bound. A planning-cost function is computed, which includes the uncertainty in the state (a trace of the covariance matrix of the EKF state estimator) in addition to the state and control cost. In a less computationally expensive approach, the method proposed by [95] uses a Convolutional Neural Network (CNN) to fuse the Unmanned Aerial Vehicle (UAV) and Unmanned Ground Vehicle (UGV) maps in a traversability-mapping scenario. It formulates an active perception module that uses conditional entropy to guide the UAV and UGV toward high-entropy paths.

### 3.5. Statistical Analysis on AC-SLAM

Table 5 summarizes the sensor types (with descriptions), SLAM methods, path-planning approaches, and type of utility functions used in AC-SLAM approaches. Most articles use (i) camera (RGB and RGBD), Lidar, and IMU sensor data for visual–inertial odometry and dense 3D point clouds as input to the SLAM system. (ii) Most implementations use pose-graph SLAM (68.7%) as compared to filter-based SLAM (32%). This preference for graph SLAM over filter based is highly encouraged as graph SLAM has many advantages [104]. Contrary to Table 2, we can infer that in AC-SLAM, graph SLAM methods are mostly deployed. (iii) Discrete sampling algorithms, which work by performing a graph search of the environment, are mostly used by A*, D*, and RRT. (iv) Entropy (37.5%), frontier information, and distance (43.7%) are deployed for the quantification of utility at goal positions. The utility preference of frontier information and distance is encouraged as it is computationally less expensive.

Table 6 elaborates on analytical, simulation, and real robot experiments along with the environment type, collaboration architecture, collaboration parameters, loop closure, and ROS framework. The information can be summarized as the following: (i) All the articles provide analytical and simulation-based results that are promising for bridging the gap between theory and simulation. (ii) The use of real robots is only 37.5%, which is very low compared to simulated robots. One of the possible reasons that real robot usage is discouraged is their inherent constraints in communication range and power. (iii) Heterogeneous real robots are used in 66.6% of the articles that use real robots. The UGV-UAV collaboration is highly encouraged as it develops new research horizons. (iv) In total, 62.5% of the articles implement loop closure, which is highly motivated by efficient localization. (v) The usage of ROS has been limited to 31.2% of articles, which is somewhat similar to the usage outlined in Table 3 and implies the declining usage of ROS in A-SLAM research.

## 4. Discussion and Future Directions

We focused on A-SLAM and AC-SLAM methods, their implementation, and their methodology applications in selected research articles. We performed extensive qualitative and quantitative analyses on numerous parameters and discussed their merits/demerits. We would like to bring into the limelight the limitations of the A-SLAM problem and future research directions in the following sections.

### 4.1. General Limitations

These limitations can be considered as open problems persisting in A-SLAM research, and we can further explain them as:Stopping criteria: Since A-SLAM is computationally expensive especially when the utility function is computed over the entire mapped area (e.g., map entropy) or in the case of TOED, the mapping of the full information matrix is required. The quantification of uncertainties from TOED may be used as an interesting stopping criterion, as discussed by [19]. The interesting approach proposed by [106] shows the qualification of uncertainty, e.g., D-optimality and the amount of explored area to stop autonomous exploration and mapping.Robust data associations: Contrary to SLAM where an internal controller is responsible for robot action and the data association (the association between measurements and corresponding landmarks) has less impact on robot actions, in A-SLAM, a robust data association guides the controller to select feature-rich positions. The qualification of these good feature/landmark positions may be difficult, especially beyond line-of-sight measurements.Dynamic environments: In A-SLAM, the nature of the environment (static or dynamic) and the characteristics of obstacles (static or dynamic) significantly influence the utility function used to plan future actions. While the majority of the A-SLAM literature primarily addresses static environments and obstacles, this focus may not align well with the complexities of real-world scenarios marked by dynamic elements. This article advocates for a broader exploration of A-SLAM in dynamic contexts, highlighting the need for adaptable and responsive robotic systems capable of thriving in real-world conditions.Simulation environment: When considering DRL-based approaches, the model training is constrained to a simulated environment, and contrary to Deep Learning approaches, an offline dataset cannot be used. The trained model may not perform optimally in real-world scenarios with high uncertainty.

### 4.2. Limitations of Existing Methods

The limitations existing with the A-SLAM and AC-SLAM methods analyzed in the previous sections of this article can be summarized as:Limited consideration of dynamic obstacles: As mentioned earlier in Section 2.2, A-SLAM involves decision and planning in unknown environments. These environments may have dynamic obstacles, as in real-world scenarios. In such a case, the SLAM algorithm must be able to detect dynamic obstacles and recompute its utility and path. In [41], the authors employed a novel approach that leverages D* [32] with negative edge weights for adaptive path planning in the presence of dynamic obstacles. Keeping into consideration the robot localization uncertainty, this method takes into account the obstacle Euclidean distance and updates the D* planner weights. The approach in [9] detects dynamic obstacles in a crowded environment, and the utility function takes into account the change in Shannon’s entropy [59] upon the detection of dynamic obstacles.High computational complexity: A-SLAM is computationally expensive, as mentioned in Section 2.1 and Section 2.2. We find only a few approaches that tackle this issue. The authors of [38] present a scenario with multiple prior topometric subgraphs, and a novel approach is introduced. It alternates between active localization and mapping and uses maximum likelihood estimation to streamline the method’s computational complexity. In an interesting approach using the methods in Section 2.3, the authors of [34] tackle the joint-entropy minimization exploration problem by introducing two versions of RRT* [31], which use distance and entropy change per distance traveled in the utility function, hence lowering the computational complexity. In AC-SLAM, the authors of [94] introduce a novel method for strengthening weak connections in the target robot’s pose graphs by the host robot. Weak connections are identified when the covariance surpasses a predefined threshold and is resolved through communication, addressing the 1-ESP problem [63]. The methods explained in Section 2.6 provide a good basis for the development of less computationally expensive A-SLAM algorithms.Real robot deployment: A-SLAM is necessary for real robot deployment because it enables robots to make decisions about where and how to select informative goal positions, adapt to changing environments, and operate effectively and autonomously in complex and dynamic environments. From the results of Section 2.7 and Section 3.5, we recall that real robots are used only in 67% and 37% of A-SLAM and AC-SLAM methods, respectively.Limited implementation of loop closure: Loop closure [107] is important in SLAM to ensure map consistency and integrity, minimize localization errors and drift, assist with global map alignment, and optimize map accuracy. It is essential for the reliable and accurate mapping and localization required in various robotic applications. In Section 2.7 and Section 3.5, we recall that loop closure is implemented in only 51% and 62.5% of A-SLAM and AC-SLAM methods, respectively. This limited use is justified by the fact that it involves the heavy computation of minimizing the localization error and exploitation (revisiting already-traversed areas of the environment), which may not be suitable for A-SLAM, which is already computationally expensive (Section 2.1 and Section 2.2).Reasoning over graph connectivity: As mentioned in Section 2.6, new methods are available to reduce the A-SLAM computational complexity, whereby debates over the SLAM pose-graph connectivity metrics and new methods for uncertainty quantification have occurred. The authors of [66] treat the underlying graph as an estimation-over-graph (EoG) [62] SLAM problem and propose a new method of computing the D-optimality criterion over the reduced weighted graph Laplacian matrix. In an AC-SLAM approach, the method presented in [94] also exploits the graph connectivity to find weak edges in the target robot pose graph and guides the host robot to localize it.Limited ROS implementation: ROS [22] is an open-source framework for robotics research. It provides a collection of libraries for sensor integration, perception, navigation, control, visualization, and analysis for many robotics platforms [108]. From Section 2.7 and Section 3.5, we can infer that ROS usage has been limited to only 45% and 31.2% of articles on A-SLAM and AC-SLAM, respectively. This limited ROS usage is related to the increased computational overhead on the A-SLAM algorithm, which decreases the real-time and performance requirements.Less usage of dynamic approaches: As discussed in Section 2.4, these approaches treat path planning in A-SLAM as an Optimal Control Problem and work on continuous planning and action spaces. They have the advantage of incorporating the robot kinematic and dynamic models into the cost function, resulting in a smooth trajectory [109] with dynamic obstacle detection. Our analysis in Section 2.7 signifies that only 11% of A-SLAM methods use this approach, mainly due to the associated computational overhead.Lack of usage of heterogeneous robots: Utilizing heterogeneous robots in AC-SLAM can enhance the mapping accuracy, improve the system robustness, increase coverage, enable multiperspective mapping, and support efficient exploration by leveraging complementary sensor modalities and capabilities among the robots. Section 2.7 shows that 66.6% of approaches use UGV and UAV collaboration for collaborative mapping [89,93,95] and information gathering [101].Managing robust communication: Robust communication implies the ability of a network to function smoothly when one or many robots/servers fail. In the case of A-SLAM, it is deduced to be a system that is immune to failure when any agent loses localization or gets out of the communication range. Most of the AC-SLAM approaches in Section 3 fail to address this issue. An interesting approach by the authors of [96] proposes a rendezvous method to manage the limited communication bandwidth by relocating robots to predefined positions when they move beyond the communication range.Communication bandwidth management: Communication bandwidth management is vital in AC-SLAM for ensuring real-time collaboration, minimizing latency, conserving resources, and optimizing cost efficiency. From the analysis in Section 3, we can infer that no AC-SLAM implementation addresses communication bandwidth management. The approach presented in [110] proposes many different visual and visual–inertial information-sharing schemes for SLAM and loop closure, which are bandwidth friendly and can be applied in AC-SLAM.

### 4.3. Future Prospects

We believe that the following areas need more investigation and may provide promising future research directions.

Detection and avoidance of dynamic obstacles: Dynamic obstacle detection and avoidance are crucial for autonomous robot navigation in unfamiliar or partially known environments. The effective management of both static and dynamic obstacle avoidance directly impacts uncertainty propagation and system entropy. In [111], the authors introduce a perception-aware Next-Best Viewpoint Planner (NBVP) [112] designed for dynamic obstacles incorporating active-loop closure. This planner employs a keypoint filtration and selection method based on the yaw angle, the number of previously detected keypoints, and the UAV’s distance to choose optimal loop-closure keypoints. Additionally, in a computationally efficient approach by [113], the authors combine multiple obstacle detectors and utilize a Kalman filter for efficient dynamic obstacle detection and tracking. For Lidar measurements, [114] proposes a method involving dynamic object segmentation and classification based on kd-nearest neighborhood search [115].Lowering computational complexity for real-time applications: As discussed earlier, the utility criterion in TOED and relative entropy computation are both computationally extensive tasks, thus limiting the real-time performance of A-SLAM. Machine learning approaches like CNNs can be used to reduce the computational overhead of loop closure in SLAM, as proposed by the authors of [116]. Leveraging the advantages of edge-cloud computing [117], the robot pose and local/global map can be estimated by utilizing the edge-cloud processing capabilities, as used by the authors of [118].Application of DRL methods: As mentioned in Section 2.4, DRL methods have the capacity to handle complex decision-making processes in continuous states and action spaces. They enable adaptive exploration–exploitation trade-offs, making them well suited for the challenges encountered in real-world A-SLAM scenarios. Deep Q networks (DQN) and double dueling (D3QN) are applications of such DRL approaches used by [13,55]. A memory-efficient solution as compared to traditional DRL approaches is presented by the authors of [119] where the robot already has a partial map of the environment inside the external memory and uses a neural-network-based navigation strategy for autonomous exploration. The method in [120] presents an interesting DRL A-SLAM method that improves exploration by incorporating the robot pose uncertainty in the reward function to favor loop closure.Advanced simulators: The use of advanced simulators is crucial for A-SLAM due to their realistic modeling, cost efficiency, and support for diverse scenarios. Commercially available simulators like AirSim [121], Carla [122], and Webots [123] provide realistic modeling of urban environments for UGV, UAV, and cars. For multirobots, MvSim [124], and for SLAM, the Virtual Reality (VR)-supported simulator proposed in [125], can be used.Multisensor fusion: Multisensor fusion enhances perception, adaptability, obstacle detection, reliability, loop closure, tracking, and localization in real-world A-SLAM. Sensor fusion is a very mature topic, and interested readers are directed to [126,127] for review articles. When using Deep Learning (DL) approaches is concerned, the authors of [128] present an actor–critic self-adopting agent for the weighing-sensors (camera, Lidar, IMU, GPS) SLAM method. An interesting method for fusing light and Lidar measurements to map and localize agents by using an Extended Kalman Filter (EKF) is presented by the authors of [129].Graph connectivity metrics: As explained in Section 2.6, a novel approach for assessing A-SLAM uncertainty is presented. This approach involves quantifying the reliability of SLAM through measures such as algebraic, degree, and tree connectivity within the pose graph. It is noteworthy that this method, as demonstrated in [64,65,66], provides a computationally efficient alternative to A-SLAM.Advance embedded design: Advanced embedded design methods are essential for real-time processing, low latency, sensor integration, robustness, real-world testing, and the overall optimized performance of A-SLAM. Referring to Table 3 and Table 6, we can infer that most approaches use commercially available robots with limited embedded processing capabilities. The authors of [130] propose an efficient Field Programmable Gate Arrays (FPGA)-based vision system for obstacle detection in real time at 30 Frames Per Second (FPS). To improve the computational capabilities for navigation tasks, the authors of [131] propose a method to add an external embedded board to the Khepra IV [132] robot.Internet of Things (IoT) and cloud computing: IoT and cloud computing offer scalable data processing, remote access, data fusion, and machine learning capabilities. The approach adopted by the authors of [133] proposes a Deep Learning (DL) model incorporated with cloud computing and IoT technologies and works with wearable glucose-level-monitoring equipment for the efficient future prediction of blood glucose levels.

## 5. Conclusions

This article focused on two emerging techniques applied in simultaneous localization and mapping technology, i.e., A-SLAM and AC-SLAM. We reviewed papers published in the past decade and collated their contributions. We started our work by recalling the SLAM problem and its formal formulation, discussing submodules and presenting methods applied for the deployment of modern active and Active Collaborative SLAM. We broadly categorized A-SLAM into four categories: geometric, dynamic, hybrid approaches, and reasoning over spectral graph connectivity depending on the trajectory-generation method, environment representation, and uncertainty quantification. We presented an extensive qualitative and quantitative analysis of the surveyed research articles and presented the research domains and methodology. For AC-SLAM, we presented the network topology and its application in collaborative localization, exploration, and trajectory-planning domains. We also performed extensive qualitative and statistical analyses of various AC-SLAM parameters. Lastly, we elaborated the limitations of the existing methods and proposed some research axes that require attention. The previous studies of [19,20] did not focus on A-SLAM problem formulation, the application of dynamic approaches, single- and multirobot statistical analysis, and providing future perspectives. Only [21] addresses these issues but does not address AC-SLAM statistical analyses and briefly comments on AC-SLAM methods and A-SLAM statistical analysis. We believe that this article offers a more-thorough exploration of the A-SLAM formulation, methods, limitations, and future prospects compared to previous works. We present an innovative and in-depth qualitative and quantitative analysis of AC-SLAM, focusing on research articles primarily from the past decade, making this review particularly valuable for emerging researchers.

## Figures and Tables

**Figure 1 sensors-23-08097-f001:**
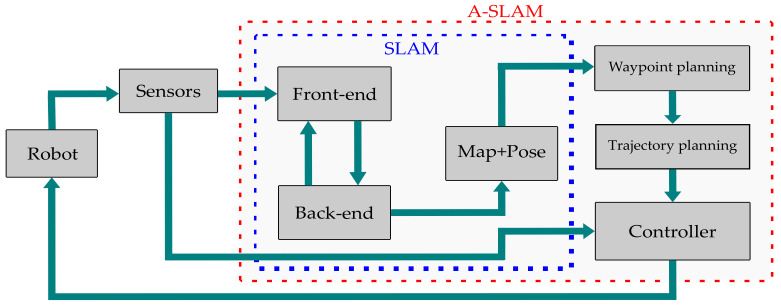
Architecture of SLAM and A-SLAM.

**Figure 2 sensors-23-08097-f002:**
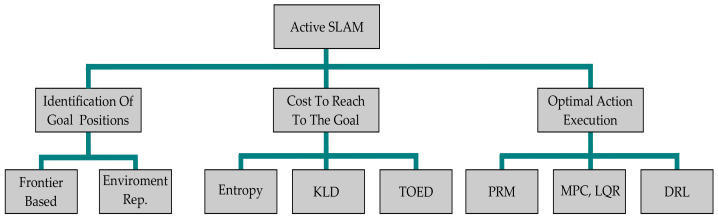
A-SLAM submodules.

**Figure 3 sensors-23-08097-f003:**
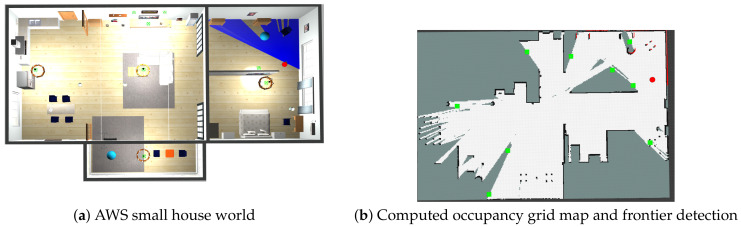
(**a**) AWS-simulated house environment: red= robot and blue = Lidar scans. (**b**) Frontier detection on the occupancy grid map: red = robot, green = detected frontiers (centroids), white = free space, gray = unknown map area, and black = obstacles [28].

**Figure 4 sensors-23-08097-f004:**
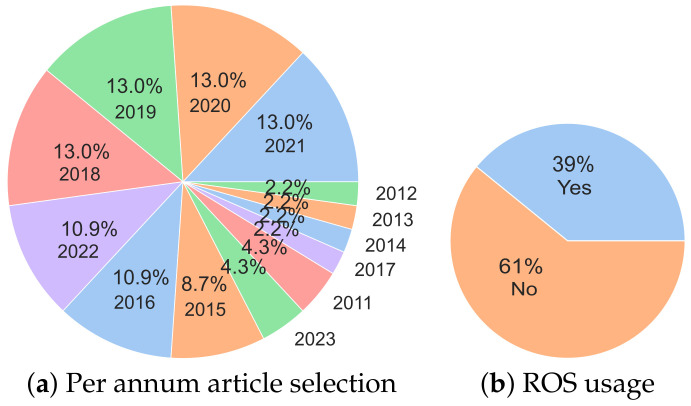
A-SLAM per annum article selection, Robot Operating System (ROS) applicability.

**Figure 5 sensors-23-08097-f005:**
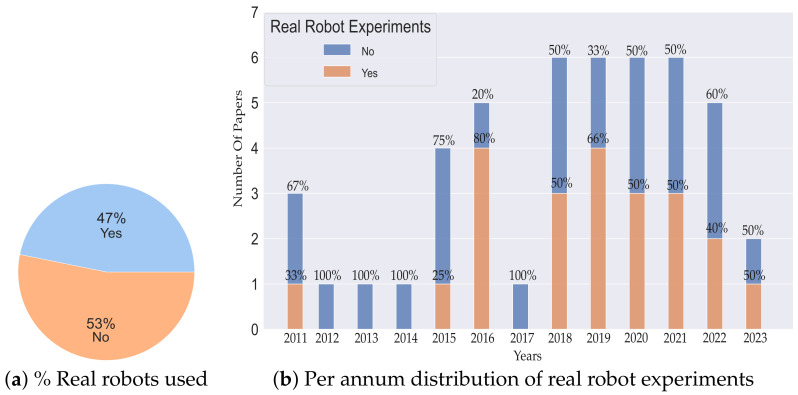
Real robot usage in A-SLAM and number of papers/articles.

**Figure 6 sensors-23-08097-f006:**
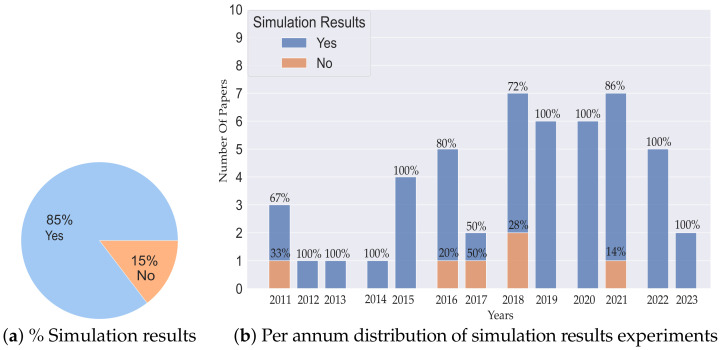
Simulation results used in A-SLAM and number of papers/articles.

**Figure 7 sensors-23-08097-f007:**
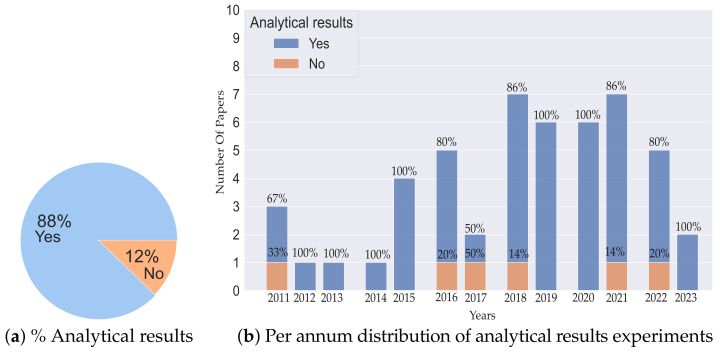
Analytical results used in A-SLAM and number of papers/articles.

**Figure 8 sensors-23-08097-f008:**
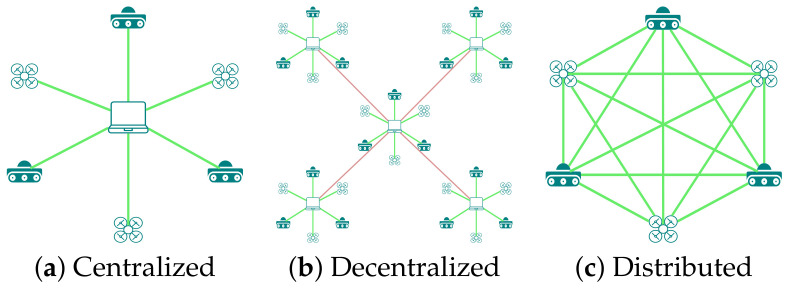
AC-SLAM network topology.

**Table 1 sensors-23-08097-t001:** Comparison of topics addressed in previous surveys.

Topics	[19]	[20]	[21]	Ours
Problem formulation	✗	✗	✓	✓
Entropy, TOED	Briefly	Briefly	✓	✓
DRL, MPC, LQR	Briefly	✗	✓	✓
Single-robot analysis	Briefly	✓	✓	✓
Single-robot stat. analysis	✗	Briefly	Briefly	✓
Multirobot methods	✗	✗	Briefly	✓
Multirobot stat. analysis	✗	✗	✗	✓
Limitations	Briefly	Briefly	✓	✓
Future perspectives	✗	✗	✓	✓

**Table 2 sensors-23-08097-t002:** A-SLAM sensors, SLAM methods, and path-planning approaches.

Article	Year	Sensors	SLAM Method	Path Planning
[7]	2017	Lidar	EKF SLAM	Active revisit path planning
[43]	2016	RGBD ^1^, Lidar ^2^, IMU ^3^, WE ^14^	ES-DSF (EKF based) ^15^	A*
[72]	2018	Lidar, RGB	Hector SLAM	Artificial potential fields
[58]	2022	Lidar ^4^, RGBD ^5^	Graph based	NMPC ^16^, A*
[73]	2019	RGB	Graph based	CAO ^20^
[6]	2016	Lidar, RGB	EKF-SLAM	Maze solver algorithm
[35]	2011	Lidar	Particle filter	Joint entropy, EMMI ^17^
[56]	2021	Lidar	ACML	-
[34]	2015	Lidar	Graph SLAM	RRT*
[39]	2015	Lidar ^6^	FastSLAM	A*, DWA ^18^
[74]	2018	Lidar	Gmapping	Sequential Monte Carlo
[8]	2020	RGB	EKF based	MPC
[46]	2019	Lidar	Graph SLAM	CPP ^19^
[75]	2016	Lidar, RGB	IEKF SLAM	Dijkstra, VSICP ^21^
[55]	2011	Lidar ^7^, RGB	Metric-based scan-matching SLAM	Reinforcement Learning
[42]	2020	Lidar, RGBD, IMU	Graph SLAM (iSAM2)	Straight line search for each hypothesis
[36]	2014	Lidar	Particle filter	Frontier-based exploration
[61]	2018	ORS ^27^	EKF SLAM	MPC
[10]	2016	RGB	Graph SLAM (GTSAM)	Bayes tree, RRT*
[76]	2018	RGBD ^8^	ORB-SLAM2	RRT*
[37]	2016	RGB	TFG SLAM	Probabilistic Road Map
[9]	2019	Lidar	Canonical scan matcher + iSAM2	Dijkstra, DOO ^22^
[49]	2012	RBS ^28^	EKF-SLAM	A*
[77]	2019	-	EKF localization	A*
[41]	2018	Lidar 2D ^9^, 3D ^10^	Graph SLAM-based ESDSF ^23^	Modified D*
[11]	2019	MBS ^29^	Graph SLAM	RRT*
[78]	2013	RGB	EKF SLAM	OCBN ^24^
[79]	2015	Lidar, IMU	Sensor-based SLAM	-
[13]	2020	RGBD, Lidar ^11^	FastSLAM	DRL ^25^
[12]	2020	RGB, IMU	Graph SLAM	RRT
[38]	2022	RGB	ORB-SLAM	RPP ^26^
[80]	2021	Lidar, IMU	RIEKF SLAM-based A-SLAM	-
[80]	2021	RGB	Object SLAM	-
[57]	2019	RGBD ^1^	ORBSLAM 2	TEB local planner
[65]	2022	Lidar	Gmapping	Deep Q learning
[26]	2020	RGBD ^12^,^13^, IMU	Graph SLAM	-
[64]	2023	Lidar	OpenKarto (g2o)	DWA ^18^
[81]	2020	Lidar	Gmapping	DDPG ^30^
[82]	2023	Lidar	Graph SLAM	A*
[66]	2021	Lidar	Open Karto (g2o)	Dijkstra
[83]	2022	Lidar ^31^	EKF SLAM	A*
[84]	2022	RGBD ^5^, IMU	ORBSLAM 3, VINS Fusion	-

^1^ Microsoft Kinect. ^2^ SICK LMS-100. ^3^ X-Sense MTI-G-700. ^4^ Hokuyo A2M8. ^5^ Intel RealSence D435i. ^6^ SLICK LMS 200. ^7^ Hokuyo URG-04LX. ^8^ Microsoft Kinect. ^9^ SICK LMS 100-10000. ^10^ Volodyne. ^11^ RpLidar A2. ^12^ Bionic eyes. ^13^ Intel RealSence T265. ^14^ Wheel Encoders. ^15^ Exactly Sparse Delayed State Filter. ^16^ Nonlinear Model Predictive Control. ^17^ Expected map mean information. ^18^ Dynamic Window Approach. ^19^ Chinese Postman Problem. ^20^ Cognitive-Based Adaptive Optimization. ^21^ Visual Servoying using successive ICP. ^22^ Dynamic obstacle avoidance. ^23^ Extremely Sparse Delayed State Filter. ^24^ Optimal-control-based navigation. ^25^ Deep Reinforcement Learning. ^26^ Rural Postman Problem. ^27^ Omnidirectional Range Sensor. ^28^ Range-Bearing sensor. ^29^ Multibeam sonar. ^30^ Deep Deterministic Policy Gradient. ^31^ LD-OEM1000.

**Table 3 sensors-23-08097-t003:** A-SLAM robot types, drive type, datasets, loop closure, ROS, map type, and utility function usage.

Articles	Robots	Drive Type	Dataset	Loop Closure	ROS	Map Type	Utility Function
[72]	CD ^6^	SS ^1^	-	-	✓	OG ^7^ and PC ^8^	FD ^9^
[58]	Robotino	OD ^3^	-	✓	✓	OG	Entropy
[73]	Survyer SVS	TD ^4^	-	-	-	OG and PC	Visual features
[6]	Khepra	DD ^2^	-	-	-	OG	Image corners
[35]	-	-	ACES, Intel Research Labs, Friburg 079	-	-	OG	Entropy
[56]	TurtleBot 2	DD	-	-	✓	OG	Particle clustering
[34]	-	-	Friburg 079	✓	-	OG	Entropy
[39]	Pioneer 3-DX	DD	-	-	✓	OG	FD
[46]	-	DD	MIT CSAIL, Intel Research Lab, AutoLab ROS	-	-	TM ^10^	D-optimality
[75]	Pioneer DX3	DD	-	-	-	TM	Entropy
[42]	-	-	-	✓	-	TM	FD
[36]	-	-	ACES, Intel Research Labs, Friburg 079	✓	-	OG	KLD
[61]	-	-	-	✓	-	TM	D-optimality
[76]	Jackal Robot	SS	-	✓	✓	OG	FD
[37]	TurtleBot	DD	-	-	✓	TM	Entropy
[9]	Pioneer 3-DX, Pepper	DD, Humanoid type	-	✓	✓	OG	D-optimality
[49]	-	-	DLR Dataset	-	-	TM	D-optimality
[41]	Clearpath Huskey	DD	-	✓	✓	TM	Distance based
[11]	Girona 500	AUV ^5^	-	-	-	TM	Entropy
[13]	TurtleBot 3	DD	-	-	-	OG	Exploration
[38]	-	-	-	✓	-	TM	Distance
[85]	-	-	-	✓	-	TM	Distance
[80]	-	-	-	✓	-	Segmented	Entropy
[57]	CD	SS	-	✓	-	PC	ORB Features
[65]	TurtleBot 3	DD	-	✓	✓	OG	D-optimality
[26]	TurtleBot 3	DD	-	-	✓	OG	Entropy
[64]	TurtleBot 3	DD	Friburg 079, CSAIL, FRH, MIT, INTEL	✓	✓	TM	D-optimality
[81]	Husarion ROSbot	SS	-	-	✓	TM	Entropy
[82]	JackalRobot, custom designed	SS, DD	-	✓	✓	3D OG	FD
[66]	TurtleBot	DD	FRH	✓	✓	OG	D-optimality
[83]	Omnidirectional robot	OD	-	-	-	OG	Distance based

^1^ Skid-steering (four-wheel drive). ^2^ Differential drive. ^3^ Omnidirectional drive (Mecanum wheels). ^4^ Traction drive. ^5^ Autonomous Underwater Vehicle. ^6^ Custom designed. ^7^ 2D occupancy grid. ^8^ 3D point cloud. ^9^ Frontier detection. ^10^ Topometric.

**Table 4 sensors-23-08097-t004:** AC-SLAM network topology and collaboration parameters.

Articles	Network Topology	Collaboration Parameters
[86]	✤ ^1^	Multirobot belief evolution by incorporating mutual observations and future measurements
[92]	◆ ^2^	Relative observation between agents
[88]	◆	Localization utility, information gain, cost of navigation
[93]	◆	Visual features, map points
[94]	✤	Weak edges in pose graphs of target agents
[95]	✤	Frontier points and map information
[96]	◆	Localization utility, information gain, cost of navigation
[89]	◆	Visual features, optimized paths
[90]	◆	Pose and map entropy, Kullback–Leibler divergence
[91]	✤	Relative pose entropy
[97]	✤	Visual features, chained localization
[87]	✤	Multirobot belief evolution by incorporating mutual observations and future measurements
[98]	◆	Frontier points and frontier-to-robot distances
[99]	◆	Frontiers and relative-position estimates
[100]	◆, ✤	Entropy and future measurements
[101]	◆, ✤	Information vector and information matrix

^1^ Centralized. ^2^ Distributed.

**Table 5 sensors-23-08097-t005:** AC-SLAM sensors, SLAM methods, path-planning approaches, and utility functions.

Papers	Years	Sensors	SLAM Method	Path Planning	Utility Function
[86]	2018	Lidar, RGB	Pose-graph SLAM	Probabilistic Road Map	Evolution of uncertainty
[92]	2020	RGBD ^1^, IMU	ORB-SLAM2	-	Subgraph info. and distance
[88]	2011	Lidar, IMU	EKF-SLAM	A*	FI ^3^, distance, evolution of uncertainty
[93]	2019	Lidar, RGBD ^1^, magnetic compass, IMU	Vision-based SLAM	FSOTP ^2^, BIT*-H [105]	FI
[94]	2020	RGB, IMU	Pose-graph SLAM	RRT	Pose-graph connectivity
[95]	2015	Lidar, RGB	Visual SLAM	-	Conditional entropy
[96]	2013	Lidar, RGB	EKF-SLAM	Frontier based	FI, distance, evolution of uncertainty
[89]	2017	Lidar, RGBD ^1^, RGB, IMU	Vision and Lidar SLAM	FSOTP ^2^, BIT*-H	FI, distance
[90]	2019	Lidar, IMU	Graph SLAM	-	Entropy
[91]	2013	Lidar, IMU	EKF SLAM	-	Relative entropy
[97]	2018	RGB	ORBSLAM2	D*	Localization uncertainty
[87]	2015	Lidar, IMU	Graph SLAM	RRT*	Localization uncertainty
[98]	2020	Lidar	Google Cartographer	-	FI, distance
[99]	2022	UWB, WiFi	Graph SLAM	-	FI, mutual distance
[100]	2015	Lidar	Information filter	-	Entropy, MI
[101]	2018	Lidar	EKF	-	Entropy

^1^ Microsoft Kinect. ^2^ Fixed-Start Open Traveling Salesman Problem. ^3^ Frontier information.

**Table 6 sensors-23-08097-t006:** AC-SLAM results, robot types, collaboration architecture, and parameters.

Papers	Analytical Results	Sim. Results	Real Robots	Env.	MR ^2^	Robot Types	Loop Closure	ROS
[86]	✓	✓	✗	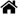 ^1^	Two	-	✓	-
[92]	✓	✓	✗	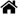	Four	-	✓	✓
[88]	✓	✓	✗	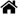	Two	-	-	-
[93]	✓	✓	✓	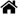	Two	UAV, UGV (custom made)	✓	✓
[94]	✓	✓	✓	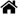	Two	UAV (custom made)	✓	-
[95]	✓	✓	✓	 ^3^	Two	UAV, UGV	-	-
[96]	✓	✓	✗	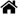	✓	-	-	-
[89]	✓	✓	✓	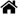	Two	UAV, UGV	✓	-
[90]	✓	✓	✗	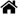	✓	-	-	-
[91]	✓	✓	✗	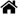	✓	-	✓	-
[97]	✓	✓	✗	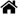	✓	-	✓	✓
[87]	✓	✓	✗	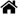	✓	-	✓	-
[98]	✓	✓	✗	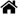	Five	-	✓	✓
[99]	✓	✓	✓	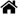	Two	UGV (TurtleBot 3)	✓	✓
[100]	✓	✓	✗	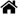	✓	-	-	-
[101]	✓	✓	✓	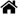	Five	UGV, UAV	-	-

^1^ Indoor environment. ^2^ Multirobot. ^3^ Outdoor environment.

## Data Availability

No new data were created or analyzed in this study. Data sharing is not applicable to this article.

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
