# Peer review of "Active SLAM: A Review on Last Decade"

_sensors, 2023, doi:10.3390/s23198097_

Round 1
Reviewer 1 Report
This manuscript is a review of the latest research on Active and Collaborative SLAM, exploring its applications in trajectory generation and control action selection. It discusses the history and development of Active SLAM, as well as its limitations and future perspectives. The file also compares different topics addressed in previous surveys and provides information on the authors' contributions and funding sources.
I am of the opinion that the manuscript could potentially attain acceptance after the implementation of minor revisions.
General comments
The introduction and the conclusion need to be improved, as it does not describe all the sections presented in the manuscript.
Some comments:
Line 401 is not clear, what does "[?]" mean?
Line 634, in Table 6, the references [101 - 104] are marked in red. Is there a specific reason for this, or is it simply an error?
Lines 669 to 678, section 3.2 should be improved.
The section pertaining to conclusions necessitates refinement in order to achieve heightened levels of clarity and impactful presentation.
Author Response
Dear Reviewer,
Thank you for your time for reviewing our paper. Your review is highly appreciated and the paper is updated accordingly.
Please find our answers in the pdf file.
Best regards
The authors

Reviewer 2 Report
1. The abstract has to highlight the contribution and novelty.
2. The abstract has to be lengthened. It is very short.
3. The abbreviations has to be expanded when firstly mentioned.
4. The keywords are related to each other. Please, try to find a diversity of keywords.
5. In Eq.(1), the authors have to explain the physical meaning of state X and the functionality of covariance matrix Ω.
6. According to Table 1, this study has the same objectives of Ref. 6.
7. The contributions have to be stated in points at the end of first section.
8. Not all blocks are explained in Figure 1. At least inside SLAM scheme.
9. The formulation of A-SLAM focused on one previous study. The authors have to conduct diversity of researches in analysis.
10. The review is descriptive and no quantitative comparison are made.
11. The authors may use the following work when addressing IoT and Cloud Computing in robot doi.org/10.3390/electronics10212719.
12. The Future Prospects has not mentioned the advanced techniques in embedded design to be as future prospective.
13. The conclusion is descriptive and it did not drew the concluded points from previous studies.
Author Response

(The authors gave the same response as above.)

Reviewer 3 Report
In this paper, a review of Active SLAM is present. The formulation, application, and methodology applied in A-SLAM for trajectory generation and control action selection using information theory and TOED based approaches have also been discussed.
Several comments are given to improve the paper quality.
1. The newest references should be added to reflect the current research status.
2. In Line 45, the introduction of the paper structure is not clear. And it is recommended to add a table of contents.
3. The limitations of existing methods should be discussed in detailed. Quantitative data is useful for reading the review.
4. The discussion about future works in this paper is not in-depth.
5. There are some typos and unclear language descriptions. The authors are suggested to check the manuscript more carefully.
Thus, a minor revision is needed.
There are some typos and unclear language descriptions. The authors are suggested to check the manuscript more carefully.
Author Response

(The authors gave the same response as above.)

Round 2
Reviewer 2 Report
There is no further comments. Thank you.